# Oocyte-Specific Knockout of Histone Lysine Demethylase KDM2a Compromises Fertility by Blocking the Development of Follicles and Oocytes

**DOI:** 10.3390/ijms231912008

**Published:** 2022-10-09

**Authors:** Xianrong Xiong, Xiaojian Zhang, Manzhen Yang, Yanjin Zhu, Hailing Yu, Xixi Fei, Fuko Mastuda, Daoliang Lan, Yan Xiong, Wei Fu, Shi Yin, Jian Li

**Affiliations:** 1Key Laboratory of Qinghai-Tibetan Plateau Animal Genetic Resource Reservation and Exploitation of Ministry of Education, Southwest Minzu University, Chengdu 610041, China; 2Key Laboratory for Animal Science of National Ethnic Affairs Commission, Southwest Minzu University, Chengdu 610041, China; 3Center for Assisted Reproduction, Sichuan Academy of Medical Science, Sichuan Provincial People’s Hospital, Chengdu 610072, China; 4Laboratory of Theriogenology, Graduate School of Agricultural and Life Sciences, The University of Tokyo, Tokyo 113-8657, Japan

**Keywords:** *Kdm2a*, folliculogenesis, oocyte, epigenetic modification, subfertility

## Abstract

The methylation status of histones plays a crucial role in many cellular processes, including follicular and oocyte development. Lysine-specific demethylase 2a (KDM2a) has been reported to be closely associated with gametogenesis and reproductive performance, but the specific function and regulatory mechanism have been poorly characterized in vivo. We found KDM2a to be highly expressed in growing follicles and oocytes of mice in this study. To elucidate the physiological role of *Kdm2a*, the zona pellucida 3-Cre (*Zp3*-*Cre*)/LoxP system was used to generate an oocyte *Kdm2a* conditional knockout (*Zp3*-*Cre*; *Kdm2a^flox/flox^*, termed *Kdm2a* cKO) model. Our results showed that the number of pups was reduced by approximately 50% in adult *Kdm2a* cKO female mice mating with wildtype males than that of the control (*Kdm2a^flox/flox^*) group. To analyze the potential causes, the ovaries of *Kdm2a* cKO mice were subjected to histological examination, and results indicated an obvious difference in follicular development between *Kdm2a* cKO and control female mice and partial arrest at the primary antral follicle stage. The GVBD and matured rates of oocytes were also compromised after conditional knockout *Kdm2a*, and the morphological abnormal oocytes increased. Furthermore, the level of 17β-estradiol of *Kdm2a* cKO mice was only 60% of that in the counterparts, and hormone sensitivity decreased as the total number of ovulated and matured oocytes decreased after superovulation. After deletion of *Kdm2a*, the patterns of H3K36me2/3 in GVBD-stage oocytes were remarkedly changed. Transcriptome sequencing showed that the mRNA expression profiles in *Kdm2a* cKO oocytes were significantly different, and numerous differentially expressed genes were involved in pathways regulating follicular and oocyte development. Taken together, these results indicated that the oocyte-specific knockout *Kdm2a* gene led to female subfertility, suggesting the crucial role of *Kdm2a* in epigenetic modification and follicular and oocyte development.

## 1. Introduction

Epigenetics play crucial roles in the establishment and maintenance of cell type specific gene expression profiles, without changing the primary DNA sequences [1,2]. Epigenetic modifications are involved in normal development and organ differentiation, including methylation, acetylation, phosphorylation, and ubiquitylation of histones [3]. These modifications are crucial in regulating cellular processes and functions such as DNA replication and repair, gene transcription, and cell cycle progression by altering chromatin condensation and DNA accessibility, thereby affecting many biological processes during development [3,4]. Among the variety of histone modifications observed during early development, changes in the methylation status of lysine residues in histone H3, lysine 4 (H3K4), lysine 9 (H3K9), lysine 20 (H3K20), lysine 27 (H3K27), and lysine 36 (H3K36) have been shown to affect reproductive physiology and germ cell development in mammals [5,6,7,8]. The patterns of methylation on these lysine residues are spatiotemporally controlled by the actions of histone methyltransferases and demethylases (KDMs) [9]. Although several KDMs, such as KDM1A, KDM3B, KDM4A, KDM4B, and KDM5A, have been extensively studied at molecular and cellular levels in in vitro cultured cells [5], the biological functions of these histone modifications in vivo remain unclear, especially on follicular growth, oogenesis, and female reproduction.

Lysine-specific demethylase 2 (KDM2) contains an F-box domain and a JmjC domain, and specifically demethylases histone H3 lysine 36 (H3K36me1/2/3) [10]. KDM2a and KDM2b, which belong to the KDM2 family, have similar structures and histone demethylase activity toward histone H3K36 [11]. Several previous studies have demonstrated the dynamic quality of histone lysine methylation during various processes in embryonic development, suggesting the critical roles for corresponding *Kdm2* during these processes [12,13]. *Kdm2b* was reported to be enriched in PRC1-repressed genes related to embryonic development, pluripotency, morphogenesis, and cellular differentiation in ESCs [14]. In addition, knockout of *Kdm2b* causes a defect in embryonic body formation, similar to the depletion of PRC1 [14,15]. Interestingly, *Kdm2b* prevents DNA hypermethylation of target CpG islands, and its loss leads to increased embryonic lethality, especially in females [15]. *Kdm2b* has also been reported to regulate the proliferation of spermatogonia and to ensure long-term sustainable spermatogenesis in mice [16]. *Kdm2a* is required for proper regulation of cell-cycle genes, and loss of *Kdm2a* leads to embryonic lethality at E10.5-E12.5 [17,18]. Furthermore, *Kdm2a* plays an essential role in embryonic development and homeostasis by regulating cell proliferation and survival [17,19,20]. The differentiation and proliferation of mesenchymal stem cells is regulated by *Kdm2a* thorough demethylating genes that encode cell cycle regulators, including p15^Ink4b^ and p27^Kip1^ [21,22,23]. However, the role of *Kdm2a* in follicular development and female reproductive function, as well as its underlying regulatory mechanisms of these events, remains mysterious.

Given the limitation in obtaining sufficient material for epigenetic analyses, the dynamic regulation of histone lysine methylation in follicular development and female reproduction only became possible due to recent progress in the establishment of a gene knockout model [24,25]. Knockout technologies have contributed fundamentally to our understanding of the cellular functions of various genes. Efficient elimination of gene expression is required to obtain a comprehensive assessment of the cellular role of a given gene. However, on the basis of cumulative evidence, more than 60% of knockout mouse strains show a prenatal lethality phenotype, including *Kdm2a* knockout [17,26]. To further study the gene functions in vivo, conditional knockout, which is a powerful tool for investigating the precise control of genetic modifications in specific tissues and stages, should be utilized. Currently, the Cre/loxP system is widely used for conditional knockout, which involves a site-specific Cre recombinase and two target sequence loxP sites. Cre is a 38 kDa site-specific DNA recombinase that recognizes a 34 bp sequence denoted loxP, leading to gene knockout only in a Cre-expressing cell or tissue [27]. To date, an increasing number of Cre-driver mice have been constructed, and they show tissue- and stage-specific expression, providing a powerful tool to generate conditional knockout individuals [28]. Thus, the aim of the present study is to investigate the effects of *Kdm2a* on oocyte growth and female fertility, and to explore the potential mechanism. To determine the function of *Kdm2a* in follicular development and female reproduction, an oocyte-specific conditional knockout (cKO) mouse was generated, deleting a floxed *Kdm2a* allele using the *Zp3*-Cre transgene, which is specifically expressed in growing oocytes [29,30]. Comparative analysis of the expression patterns of *Kdm2a* in cKO and control (*Kdm2a^flox/flox^*) was conducted to establish dynamic expression profiles during follicular development and detect the effects of *Kdm2a* on follicular growth, oocyte meiotic maturation, female reproductive function, and patterns of genes expression and H3K36me1/2/3. This study is the first to demonstrate that *Kdm2a* is highly expressed in oocytes and crucial for oocyte and follicular development, possibly because of its role in regulating epigenetic modification and gene transcription.

## 2. Results

### 2.1. Confirmation of Kdm2a Deletion

To study the physiological roles of *Kdm2a* during oocyte and follicular development in vivo, we used a mutant of *Kdm2a* floxed mice bred with *Zp3*-Cre transgenic mice to generate the *Kdm2a* cKO genotype. The genotypes of the litters were selected for the experiment, and the results are shown in Figure 1B,C. The knockout of *Kdm2a* was confirmed by sequencing, Southern blot analysis, and RT-PCR using DNA from ear biopsies of *Kdm2a* cKO and control females (*Kdm2a^flox/flox^*). These results indicated that our study successfully constructed a *Kdm2a* conditional knockout model.

### 2.2. Expression and Subcellular Localization of Kdm2a

We examined the expression profiles and subcellular localization of *Kdm2a* during oogenesis and folliculogenesis. As shown in Figure 2, the IHC results revealed that *Kdm2a* was ubiquitously expressed in oocytes and surrounding granulosa cells of primordial follicles, primary follicles, preantral follicles, and antral follicles. Although the exact mechanism by which *Kdm2a* regulates the growth of follicles remains unclear, these data suggested a potential role for *Kdm2a* during oocyte and follicular development. As shown in Figure 3, *Kdm2a* was expressed during all stages of oocyte meiotic maturation, including GV-, GVBD-, and MII-stages, and the expression levels dynamically changed. To study the subcellular specific distribution of *Kdm2a*, oocytes at different stages were detected by immunofluorescence. As shown in Figure 3A, *Kdm2a* was mostly localized in the cytoplasm, and significantly higher in MII-stage oocytes than in GV- and GVBD-stage oocytes (Figure 3B, *p* < 0.05). Furthermore, the dynamic expression of *Kdm2a* during oocyte meiotic maturation was detected by Western blot and RT-qPCR (Figure 3C–E). Although the protein of KDM2a showed a downward trend during oocyte meiosis, the mRNA of *Kdm2a* was continuously expressed in oocyte from GV-stage to MII-stage, with a high expression at MII-stage, and the lowest in GV-stage (*p* < 0.05). Thus, *Kdm2a* was dynamically expressed and played a crucial role during oocyte meiotic progression and/or subsequent development. Furthermore, in follicles and oocytes of *Kdm2a* cKO, *Kdm2a* was hardly expressed during meiosis, and the protein level of KDM2a was nearly undetectable, which indicated that *Kdm2a* was effectively knocked out in this study.

### 2.3. Decrease the Fecundity of Kdm2a cKO Female Mice

The *Kdm2a* cKO mice grew normally into adults as the control group. To investigate the effect of maternal *Kdm2a* on female reproductive performance, six female *Kdm2a* cKO mice were paired with wildtype (WT) male mice at a 1:1 ratio in each cage for 4 months, producing a total of 91 pups (15.2 pups/breeding pair). The body weights and bilateral ovarian weight were not significantly different between the *Kdm2a* cKO and control group (Appendix A). However, the same breeding paradigm of control (*Kdm2a^flox/flox^*) female and WT male mice produced a total of 175 pups (29.2 pups/breeding pair, Appendix A). Further analysis of the breeding data revealed that each breeding pair with the female *Kdm2a* cKO produced 3.2 litters in average, which was not significantly different from the control female mice (*p* > 0.05). However, the average number of pups per litter of the female *Kdm2a* cKO group was obviously fewer than that of the control group (4.8 ± 1.2 vs. 9.7 ± 0.9, *p* < 0.05), indicating a reduction of about 50% (Appendix A, Figure 4E and Appendix A). These results revealed that *Kdm2a* was required for normal female reproductive function, and loss of *Kdm2a* compromised female fertility and was mainly reflected by the reduced number of pups.

### 2.4. Effects of Kdm2a on Follicular Development

To further investigate how *Kdm2a* affects the female fertility, the composition of follicles was analyzed in ovaries at different ages (Figure 4). Ovaries from 4-week-old *Kdm2a* cKO mice exhibited no obvious abnormality in morphology and histology of each follicle stage compared with their counterparts, except antral follicle (Figure 4B). However, the ovaries of 6-week-old *Kdm2a* cKO had fewer preantral follicles and antral follicles than those of the control group (*p* < 0.05, Figure 4C). Furthermore, the numbers of preantral follicles and antral follicles in 8-week-old *Kdm2a* cKO mice considerably decreased than those of their counterparts (*p* < 0.05, Figure 4D). These results suggested that loss of *Kdm2a* was progressively detrimental to follicular development in *Kdm2a* cKO mice with aging, and partially arrested at the primary follicle stage. For further confirmation of the effects of *Kdm2a* on follicular development, 6-week-old *Kdm2a* cKO mice were subjected to PMSG to evaluate oocyte development. However, the quantity of cumulus oocyte complex (COCs) of *Kdm2a* cKO was significantly lower than that of the control, accompanied with cytoplasmic damage (Figure 5A). After in vitro culture, we measured the germinal vesicle breakdown (GVBD) and first polar body formation. The results indicated that GVBD of *Kdm2a* cKO oocytes was significantly lower than that of the control group (*p* < 0.05, Figure 5B). Meanwhile, most of *Kdm2a* cKO oocytes were arrested at GVBD-stage and failed to extrude the first polar body (*p* < 0.05, Figure 5C). Altogether, these data suggested that the subfertility of *Kdm2a* cKO female was due to abnormal follicular and oocyte development.

### 2.5. Kdm2a cKO Females Exhibited Decreased Levels of Estradiol

The female reproductive function is tightly regulated by ovarian steroid hormones including estradiol and progesterone. In 4-week-old mice, the average concentration of 17β-estradiol in the *Kdm2a* cKO females was only 60% of that in their counterparts (*p* < 0.05). After synchronized estrous cycles by PMSG and hCG treatments, the levels of 17β-estradiol were also significantly lower in the *Kdm2a* cKO mice than in the control at 24 h post-PMSG treatment and 24 h post-hCG treatment (Figure 6A). However, the progesterone concentration showed no significant difference between these groups (Figure 6B). To further examine the effects of knockdown of *kdm2a* on hormone sensitivity, we compared the number of ovulated oocytes after superovulation. The results showed that the total ovulated oocytes of the *Kdm2a* cKO group were significantly lower than that of the control group (*p* < 0.05), and most of the ovulated oocytes had no first polar body (Figure 6C), which indicated that the hormone sensitivity of *Kdm2a* cKO female remarkably decreased and corresponding oocytes were blocked at the GV- and/or GVBD-stage.

### 2.6. Kdm2a cKO Increased the Level of H3K36me2/3 during Oocyte Growth

To understand the functions of *Kdm2a* on the methylation levels during oocyte development, we examined the profiles of H3K36me1/2/3 by immunofluorescence staining. The levels of H3K36me1/2/3 were dynamically changed during oocyte growth. The results showed that the fluorescence intensity of H3K36me1 in each stage was comparable between the control and *Kdm2a* cKO oocytes (*p* > 0.05), but the levels of H3K36me2/3 in GVBD-stage oocytes of *Kdm2a* cKO were significantly higher than that in their counterparts (*p* < 0.05, Figure 7). This trend might be related to the oocyte maturation rate because the extruded first polar body rate of oocytes in *Kdm2a* cKO mice was less than that of the control group. These results suggested that *Kdm2a* was crucial for maintaining the development of follicles and oocytes, and knockout of *Kdm2a* was detrimental to the developmental potential of oocytes. 

### 2.7. Loss of Kdm2a Changed the mRNA Expression Pattern in Oocytes

To explore the potential mechanism of *Kdm2a* on follicular and oocyte development, RNA transcriptome sequencing of GV- and MII-stage oocytes from 4-week-old mice was performed (Figure 8). In general, a total of 267 genes exhibited significant differential expression in GV-stage oocytes between *Kdm2a* cKO and the control group, including 81 upregulated and 186 downregulated genes. There were also 117 upregulated genes and 65 downregulated genes in MII-stage oocytes of *Kdm2a* cKO compared with the control group (Figure 8A–C). GO analysis of the differentially expressed genes indicated that they were enriched in many biological processes, such as meiosis, apoptosis, cellular proliferation, and multiple other functions (Figure 8D). KEGG analysis also indicated that these differential genes were involved in important pathways for regulating follicular and oocyte development, such as PI3K-Akt, TGF-β, MAPK, and P53 signaling pathways (Figure 8E). All these results revealed that *Kdm2a* participated in many signaling pathways to regulate the development of follicles and oocytes involving multiple biological process.

Interestingly, numerous follicular and oocyte development-related genes were found among differentially expressed genes, such as anti-Mullerian hormone (*Amh*), bone morphogenetic protein 15 (*Bmp15*), and follicle-stimulating hormone receptor (*Fshr*). Therefore, RT-qPCR was applied to detect the mRNA expression of these genes in GV-stage oocytes after loss of *Kdm2a*. The mRNA levels of *Gdf9*, *Inha*, and *Bmp15* were significantly decreased in oocytes of *Kdm2a* cKO compared with those of the control (Figure 9). Insufficient expression of *Gdf9* and *Bmp15* in oocytes likely contributed to the arrest of oocyte growth in *Kdm2a* cKO mice, and the shortage of *Inha* acted as a barrier in follicular development. The mRNA levels of *Amh, Star*, and *Lhcgr* were significantly increased in oocytes of *Kdm2a* cKO mice (*p* < 0.05). Thus, the expression changes of these genes might mediate the effects of *Kdm2a* deletion on follicular and oocyte development. In addition, the relative expression of the *Kdm2b* gene was found to be prominently upregulated in *Kdm2a* cKO oocytes compared with the control oocytes (*p* < 0.05). This might be an attempt to compensate for the loss of *Kdm2a* in oocytes. These differentially expressed genes might mediate the effects of *Kdm2a* knockout on the development of follicles and oocyte

## 3. Discussion

Although the activity and specific function of *Kdm2a* have been partially studied [3,11,17], its physiological functions, especially in reproduction, remain unknown. In this study, we generated cKO mice of oocyte specific lacking *Kdm2a*, a Jumonji C-domain-containing histone demethylase, to check the biological function in female reproduction and corresponding mechanism. The main findings presented in this study demonstrated that *Kdm2a* cKO mice exhibited low female reproductive ability, accompanied with the arrest of follicular and oocyte development. Moreover, the patterns of H3K36me2/3 and mRNA expression were changed in oocytes after conditional knockout *Kdm2a*. Thus, we showed for the first time that *Kdm2a* played a critical role in female reproduction by regulating in vivo hormone levels, gene expression, and epigenetic modification during folliculogenesis and oogenesis.

In this study, our results demonstrated that the pups’ number of *Kdm2a* cKO female decreased by approximately 50% than that of the counterparts. Although being in different stages of the menstrual cycle has a certain impact on reproduction, we inferred that this result was caused by the abnormal development of follicles and oocytes after *Kdm2a* deletion. Folliculogenesis and oogenesis involve highly dynamic changes in morphogenesis, chromatin structure, and gene transcription [31,32], and these events in an orderly manner lead to matured oocyte ovulation and are necessary to regulate embryo development [33]. This entire process is regulated by numerous factors, including epigenetic modification. To date, the role of epigenetic mechanisms in folliculogenesis has been partially investigated, such as the biochemical modification of DNA and histones. In this study, we first examined the expression pattern of *Kdm2a* during follicular development to explore whether it plays a role in the female germ-line. The results indicated that *Kdm2a* is continuously expressed during follicular development and mainly localized in oocytes and surrounding granulosa cells. Previous study indicated that *Kdm1b* is required to establish maternal genomic imprinting, and conditional deletion of *Kdm1b* in growing oocytes results in the precocious resumption of meiosis and spindle, and chromosomal abnormalities [34]. Knockdown of *Kdm3b* decreases the quantity of pups and litters [35]. In addition, the literature has reported that *Kdm6a*-deficient PGCs show aberrant epigenetic reprogramming in vivo, which leads to germline transmission failure in mouse chimeras generated from *Kdm6a* KO mESCs [36]. Consistent with these studies, we found that the expression of *Kdm2a* in oocytes drastically changed during meiotic maturation, and loss of *Kdm2a* decreased the quality of oocytes and partial arrest at GV-/GVBD-stage. Moreover, our results revealed that *Kdm2a* showed dynamic expression throughout follicular development, and the loss of *Kdm2a* led to follicular arrest at the primary follicle stage. All these data suggested a potential role for *Kdm2a* during folliculogenesis and oogenesis progression, but the specific mechanism needs further study. 

The results observed from ovarian steroid hormone assays may help explain the causes responsible for the decreased reproductive capacity of *Kdm2a* cKO females. 17β-Estradiol is a steroid hormone derived mainly from the ovaries and it is required for both female and male reproductive functions [35,37]. Our results showed that the 17β-estradiol concentration was markedly reduced in *Kdm2a* cKO mice, but no significant change in progesterone concentration was found. These results demonstrated that *Kdm2a* is required for maintaining normal circulating levels of 17β-estradiol. Consistent with the finding reported by Liu and his colleagues, knockdown of *Kdm3b* caused the circulating levels of 17β-estradiol to significantly decrease in female and male mice [35]. Additionally, *Kdm3b* deficiency did not affect the expression of 17β-estradiol related-genes, which are involved in estrogen synthesis. In this study, the regulatory mechanisms responsible for the decreased estradiol in the female *Kdm2a* cKO mice need further research. In the future, we will further analyze the relationship between *Kdm2a* and estrogen and determine the interrelation between the change in estrogen level and reproductive competence. Fertility and oocyte maturation are also closely related to hormone sensitivity. The development of follicles and oocytes is tightly controlled by hormone levels in the body and sensitivity of receptors to hormones [38,39]. Thus, we performed superovulation and found that the total number of ovulated and matured oocytes sharply declined in the absence of *Kdm2a*, indicating that *Kdm2a* might be associated with the sensitivity of hormones to regulate the developmental process of follicles and oocytes. Our further results indicated that the cytoplasmic quality of oocytes was also affected after *Kdm2a* knockdown, and this might be the partial reason for the fewer pups. Together with previous studies, we revealed the roles of *Kdm2a* in hormonal changes and corresponding developmental dynamics of follicles and oocytes.

To explore the potential mechanism of *Kdm2a* on oocyte meiotic maturation, we examined the patterns of epigenetic modification and gene expression. In mammals, an appropriate H3K36 methylation pattern has been shown to play essential roles in cellular function and reproduction. KDMs, a family of histone demethyltransferases, are required for the regulation of methylation level. Tsukada and his coworkers reported that *Kdm2a* preferentially demethylates H3K36me2 in 293T cells [40]. Similar results have found that *Kdm2a* only demethylates H3K36me1/2/3 but not other methylates such as H3K9me1/2/3 and H3K27me [41,42]. Overexpression of *Kdm2a* in 293T cells significantly reduced the level of H3K36me2, whereas ectopic expression of *Kdm2b* has no effects on the expression of H3K36me2 [43]. However, the expression of *Kdm2b* in mouse embryonic fibroblasts significantly reduced the level of H3K36me2 [44]. These studies indicated that H3K36me is a common target for *Kdm2a* and *Kdm2b* in mammalian cells. In the present study, knocking down *Kdm2a* significantly increased the expression level of H3K36me2/3 in GVBD-stage oocytes. Knockout of *Kdm2a* possibly increased H3K36me2/3 levels and disturbed the expression of genes required for chromatin condensation and oogenesis and maturation, which blocked oocyte meiotic maturation and caused female subfertility. Thus, alterations in epigenetic modification should affect the related gene expression patterns of oocyte maturation and reproduction, which may serve as a molecular basis for the subfertility phenotype of *Kdm2a* cKO mice. In addition, the results of transcriptome sequencing revealed many differential expression genes in GV- and MII-stages of *Kdm2a* cKO oocytes. These genes are involved in key pathways related to oocyte meiosis, such as PI3K, TGF-β, and MAPK [45,46]. Therefore, differentially expressed genes respond to oocyte developmental potential after *Kdm2a* deletion. Given the difficulty in identifying the specific target genes and pathways responsible for the reproductive functions of female *Kdm2a* cKO mice, additional studies are necessary to ascertain the *Kdm2a*-regulated genes and their underlying molecular mechanisms that regulate and control female reproductive function.

In summary, this study is the first to provide new insight into *Kdm2a* and the regulatory role during follicular and oocyte development. Disruption of *Kdm2a* function decreases estradiol secretion and changes the pattern of H3K36me2/3, which compromises female reproductive performance. However, this study has several limitations, including whether *Kdm2a* influences embryo development during preimplantation, and implantation was not investigated. Furthermore, investigations of *Kdm2a* are needed to advance our understanding of their folliculogenesis mechanism.

## 4. Materials and Methods

### 4.1. Generation of Kdm2a-Deficient Mice

All mice were maintained in strict accordance with the policies of the Southwest Minzu University Animal Care and Use Committee (approval number: 2020A017). Mice were housed in 12 h alternating light/dark cycles, with free access to water and food. The exon 6 of *Kdm2a* was flanked by loxP Cre recombinase target sequences and this allele was termed *Kdm2a* (Figure 1A). The *Kdm2a* knock-in targeting vector was constructed by the bacterial artificial chromosome recombineering technique, and then electroporated into the ES cell line TC-1. Chimeric mice derived from an ES clone was used to generate F1 offspring bearing the mutant alleles, and female *Kdm2a^flox/flox^* was further crossed with *Zp3*-Cre transgenic homozygous male mice (Tg(*Zp3*-Cre)93Knw, MGI: 2176187). In this study, mutant females had the following genotypes: *Zp3*-Cre;*Kdm2a^flox/flox^* (*Kdm2a* cKO) or *Zp3*-Cre;*Kdm2a^flox/wt^*. All *Kdm2a* mutant mice were maintained in a C57BL/6J genetic background. Deletion of the *Kdm2a* gene in oocytes was confirmed by reverse transcription polymerase chain reaction (RT-PCR). Primers and PCR conditions are shown in the Appendix A.

### 4.2. Assessment of Female Reproduction

To define the genetic role of *Kdm2a* in female reproductive function, 6-week-old *Kdm2a* cKO and control (*Kdm2a^flox/flox^*) female mice were mated with 8-week-old wild type (WT) male mice for 4 months (1:1 in each cage) to evaluate fertility and reproductive capability, respectively. The number of offspring from each female was recorded after birth. 

### 4.3. Ovarian HE Staining and Immunohistochemistry (IHC)

Ovaries used for histological analysis were collected from female mice at different stages (4-week-old, 6-week-old, and 8-week-old) and fixed in 4% paraformaldehyde overnight at 4 °C. HE staining and IHC were conducted according to routine protocols with minor modification [47]. In brief, after deparaffinization and rehydration, ovaries were cut into 5 µm-thick serial sections followed by staining with hematoxylin and eosin solution (H&E). Follicles in every fifth section of the whole ovary were counted and classified as described in a previous study [48]. The follicles were classified and counted according to morphological characteristics. The primordial follicle was defined as a single layer of pregranulosa cells surrounding the oocyte, the primary follicle was identified by a single layer of granulosa cells, the preantral follicle was characterized by multiple layers of granulosa cells and exhibited no antrum, and the antral follicle was identified by the presence of an antral cavity. The cumulative number of follicles was calculated by multiplying the number of follicles visible in the nucleus of oocytes.

For IHC analysis, 5 µm sections were deparaffinized and rehydrated in a graded series of ethanol to water, followed by hydratipn with an ethanol gradient (100–70%) as described in a previous study [49]. Antigen retrieval was performed with 10 mM sodium citrate (pH 6.0), permeabilized with 0.2% TritonX-100 in PBS and incubated with the primary *Kdm2a* antibody (Abcam, ab191387, 1:500) overnight at 4 °C. The negative group was incubated with PBS in place of the primary antibody. After washing with PBS, a biotinylated secondary antibody was incubated for 45 min, followed by 3, 3’ -diaminobenzidine (DAB) and hematoxylin staining. Ovaries from no less than three mice of each genotype were used for the analysis. Images were obtained using a Zeiss LSM800 confocal microscope.

### 4.4. Oocyte Collection and In Vitro Culture

To acquire a sufficient number of oocytes, 7.5 IU of pregnant mare’s serum gonadotropin (PMSG, Ningbo Pharmacy Factory, China) was injected into a 6-week-old female, and the cumulus-oocyte-complexes (COCs) were obtained after 44–46 h by puncturing ovarian antral follicles as described in a previous study [50]. Finally, GV-stage COCs were cultured in an M16 medium (Sigma Chemical Co., St. Louis, MO, USA) in a humidified atmosphere of 5% CO_2_ at 37.5 °C for 5 and 12 h to count GVBD- and MII-stage oocytes, respectively. The matured oocytes were identified by the presence of the first polar body (PBI) under a stereomicroscope.

### 4.5. Detection of Hormone Levels and Evaluation of Hormone Sensitivity

Serum was prepared from 4-week-old control and *Kdm2a* cKO mice by periorbital puncture. The concentrations of 17β-estradiol and progesterone in serum were measured using ELISA kits ml058533 and ml057778 (Mlbio, Shanghai, China) following the manufacturer’s instruction [51]. 

To evaluate the sensitivity of hormone, mice were injected with 7.5 IU PMSG, followed by 7.5 IU of human chorionic gonadotropin (hCG, Ningbo Pharmacy Factory, China) after 48 h. At 14 h after hCG treatment, the mice were sacrificed and COCs were recovered from oviductal ampullae. The COCs were treated with hyaluronidase (1 g/L) in an M2 medium, and the total number of oocytes and extrusion of PBI were counted under a stereomicroscope. 

### 4.6. Immunofluorescence Assay

Oocytes were fixed for 1 h in 4% formaldehyde in a phosphate buffered saline (PBS, pH 7.4) at room temperature (RT). After permeabilization with 0.2% Triton X-100 for 20 min, oocytes were incubated with primary antibodies diluted in blocking solution overnight at 4 °C, including *Kdm2a* (Abcam, ab191387, 1:500), H3K36me1 (Abcam, ab176920, 1:1000), H3K36me2 (Abcam, ab176921, 1:1000), and H3K36me3 (Abcam, ab9050, 1:1000). After three washes with PBS, oocytes were incubated with corresponding secondary antibody at RT for 30 min, and stained with propidium iodide (PI, 1 μg/mL in PBS) for 5 min. Labeled oocytes were mounted on slides and obtained images using a Zeiss LSM800 confocal microscope (Carl Zeiss, Germany) with the same parameters. The ZEN imaging and ImageJ software were used to analyze the images.

### 4.7. RNA Library Construction and Sequencing

Oocytes used for single cell RNA-Seq transcriptomic analysis were collected from 4-week-old mice as described previously [52] with minor modifications. In brief, 10 denuded oocytes of each genotype (*Kdm2a^flox/flox^* and *Kdm2a* cKO) and stage (GV-stage and MII-stage) were collected and then added into lysis buffer in 0.2 mL tubes. The quantity and integrity of RNA were assessed by a 2100 Bioanalyzer and RNA 6000 Pico Kit (Agilent Technologies, Palo Alto, CA, USA). cDNA synthesis and RNAseq library were performed by SMARTer Ultra Low Input RNA for Illumina Sequencing-HV kits (Takara Bio Inc., Otsu, Japan) according to the manufacturer’s instructions. After amplification, and nucleotides were sequenced on a HiSeq 2500 platform (BGI gene Inc., Shenzhen, China).

The low-quality bases and adaptor sequences was removed using Seqyclean software with default parameters. The filtered reads were aligned to the mouse reference (GRCm38) genome by Hisat2 software [53]. Mapped reads and transcripts were processed at the gene level to account for the number of reads per gene in each sample. Differentially expressed genes between *Kdm2a^flox/flox^* and *Kdm2a cKO* were considered significant using the package of edgeR at a false discovery rate (FDR) -adjusted *p*-value (p. adjust) < 0.05 and |log_2_FoldChange| > 1.

### 4.8. RNA Extraction and RT-qPCR

RNA extraction from oocytes (20 oocytes/sample) using the Cells-to-Signal^TM^ Kit (Invitrogen, Carlsbad, CA, USA) followed the protocol of the manufacturer as previously described [54]. Briefly, each group of oocytes were washed twice with PBS and transferred to 10 μL Lysis Buffer. The sample was gently shaken for 3 min to lyse the oocytes. The isolated RNA was immediately used for reverse transcription with the cDNA synthesis kit (Takara, Dalian, China) according to the manufacturer’s instructions, and then used as templates for PCR analysis. Gene transcripts were determined via CFX96 Real-Time PCR as described previously [55], with minor modifications. Briefly, 15 µL of RT-qPCR reaction includes 1 µL of forward and reverse primers, 8 µL of SYBR Green Premix, 2 µL of sample cDNA, and 4 µL of ddH_2_O. PCR primers sequences were designed for intron-crossing using the Primer 5.0 software as shown in Appendix A, and the specificity of the amplified fragment was confirmed by dissociation curve and gel electrophoresis. Samples were run in duplicates, and the standard curve method was used to determine the abundance of mRNA for each gene. Relative mRNA expression was normalized to the mean abundance of the endogenous control gene beta-actin (*Actb*). The relative mRNA expression level against *Actb* was calculated using the 2^−ΔΔCt^ method. The experiments were repeated at least three times.

### 4.9. Western Blot

For Western blot analysis, 150 oocytes from each group were used as previously described [56]. Briefly, the antibodies of anti-Kdm2a (Abcam, ab191387, 1:1000) and anti-ACTB (Abcam, ab26273, 1:1000) were incubated with the blotted PVDF membranes overnight at 4 °C. After washing, the blot was incubated with an HRP-conjugated anti-rabbit antibody (bs-40295G-HRP) for 2 h at RT. Thereafter, a chemiluminescence detection system (iBright CL100, Thermo, Carlsbad, CA, USA) was used for quantitating the immunoreactive bands. The relative levels of proteins were normalized to the expression of ACTB.

### 4.10. Statistical Analysis

All experiments were repeated at least three times. Relative fluorescence signals were compared between the control and experimental groups using nonparametric tests (Mann-Whitney U-test). Results were presented as means ± SEM, and *p* < 0.05 was considered statistically significant. The fold change in mRNA expression was calculated using the mean expression values of three replicates from each stage of development and treatment.

## Figures and Tables

**Figure 1 ijms-23-12008-f001:**
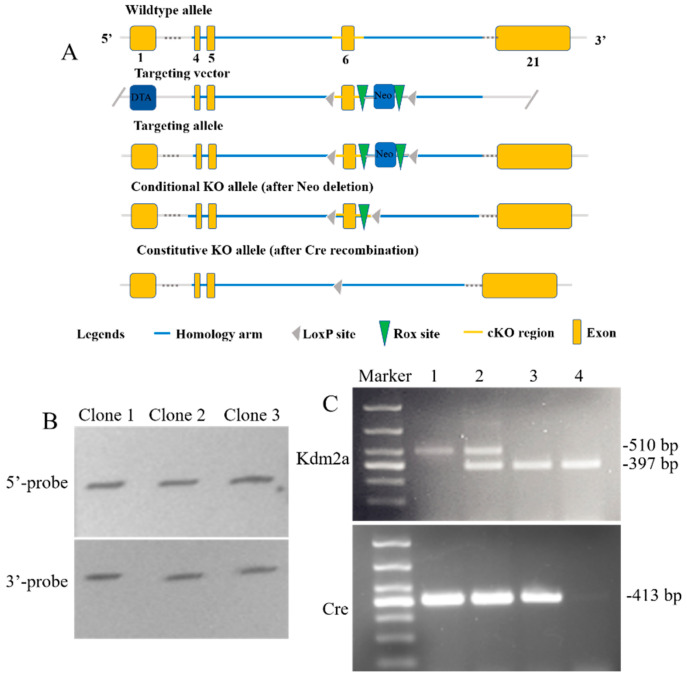
Generation of *Kdm2a* conditional knockout mice and confirmation. (**A**) The gene targeting strategy. The *Kdm2a* genomic locus, targeting vector, and targeted allele are sketched. (**B**) Southern blot analyses of *Kdm2a* floxed ESC lines with 5′ -probe and 3′ -probe. Representative results from three independent clones (*Kdm2a^flox/flox^*) were presented. The 5′ probe and the 3′ probe represent the targeted alleles as indicated. (**C**) Oocyte-specific deletion of the *Kdm2a* gene in genotyping of the litters was confirmed. M: DNA Marker, line 1: *Kdm2a^flox/flox^*; *Zp3*-Cre (cKO), line 2: *Kdm2a^flox/wt^*; *Zp3*-Cre, line 3: *Kdm2a^wt/wt^*; *Zp3*-Cre, line 4: *Kdm2a^wt/wt^*.

**Figure 2 ijms-23-12008-f002:**
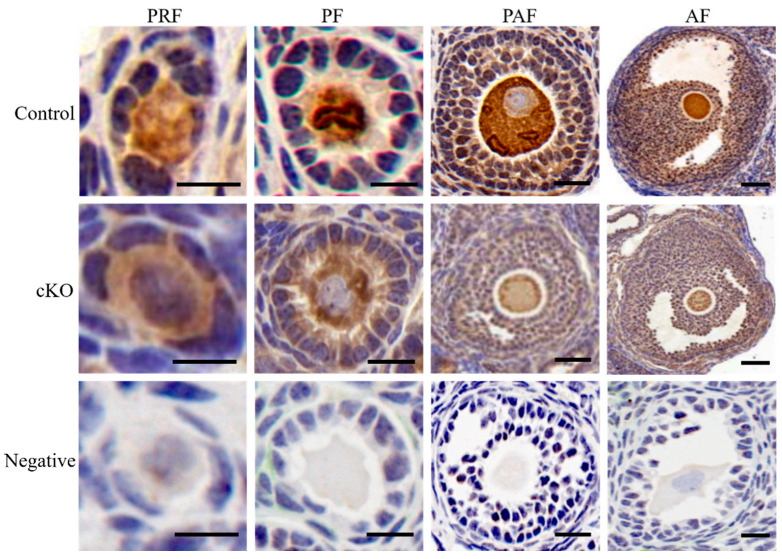
The expression of *Kdm2a* in growing follicles. Immunohistochemical (IHC) analysis showed the levels of *Kdm2a* in primordial follicles (PRF), primary follicles (PF), preantral follicles (PAF), and antral follicles (AF). Ovaries from 6-week-old control (*Kdm2a^flox/flox^*) and *Kdm2a* cKO (cKO) mice were stained with anti-*Kdm2a* and then counterstained with haematoxylin. Scale bars, 70 μm.

**Figure 3 ijms-23-12008-f003:**
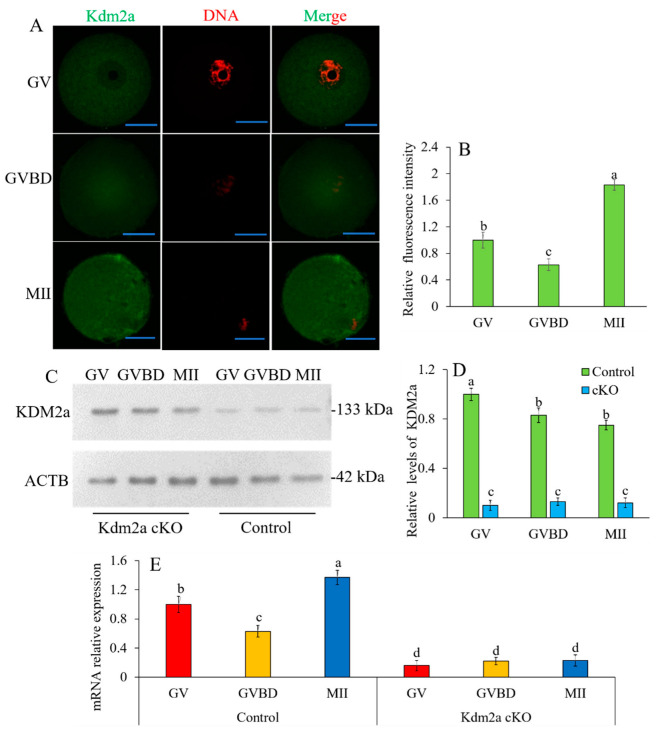
Subcellular localization and expression of *Kdm2a* during oocyte meiotic maturation. (**A**,**B**) Immunofluorescence analysis was utilized to track the spatial profile of KDM2a in GV, GVBD, and MII stages oocytes. Oocyte were labeled with anti-KDM2a followed by appropriate Alexa Fluor-conjugated secondary antibodies (green) and were counterstained with the nuclear stain PI (red). (**C**,**D**) Protein levels of KDM2a in oocytes at GV, GVBD, and MII stages were examined by western blot. (**E**) RT-qPCR analysis of *Kdm2a* mRNA relative expression levels in oocytes. The different superscript letter showed significant difference (*p* < 0.05). Scale bars, 50 μm.

**Figure 4 ijms-23-12008-f004:**
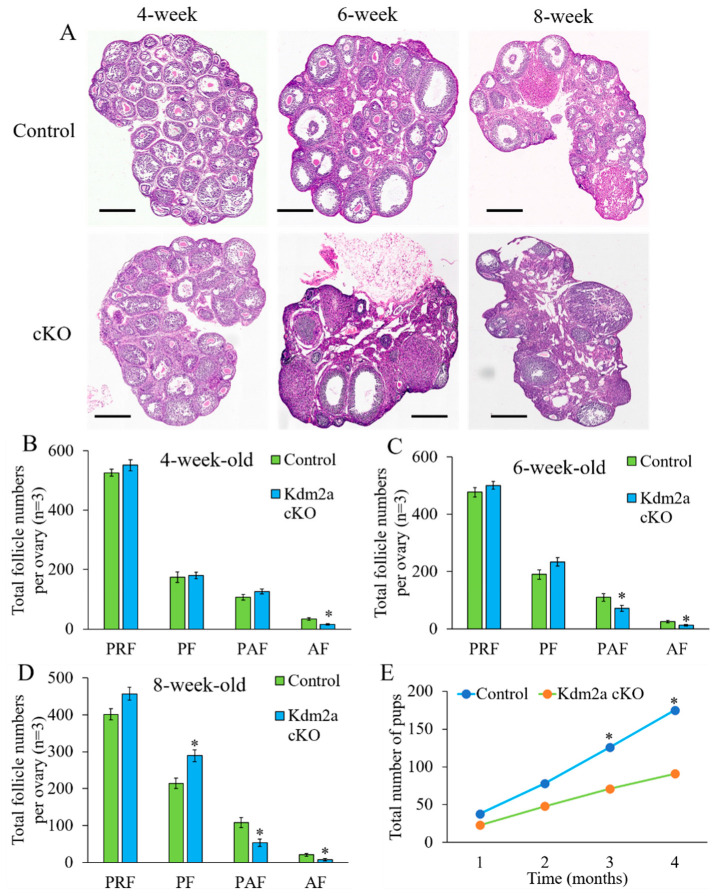
*Kdm2a* cKO ovaries display some defects in follicle development. (**A**) HE staining of 5 μm cross sections of *Kdm2a* cKO ovary and control (*Kdm2a^flox/flox^*) ovary. Scale bar, 25 mm. (**B**–**D**) Proportion of primordial follicle (PRF), primary follicle (PF), preantral follicle (PAF), and antral follicle (AF). The difference between the control and *Kdm2a* cKO was observed according to two-way ANOVA test. (**E**) The total number of pups produced from control and *Kdm2a* cKO group (n = 6) in four months. * Significant difference (*p* < 0.05). Scale bars, 50 μm.

**Figure 5 ijms-23-12008-f005:**
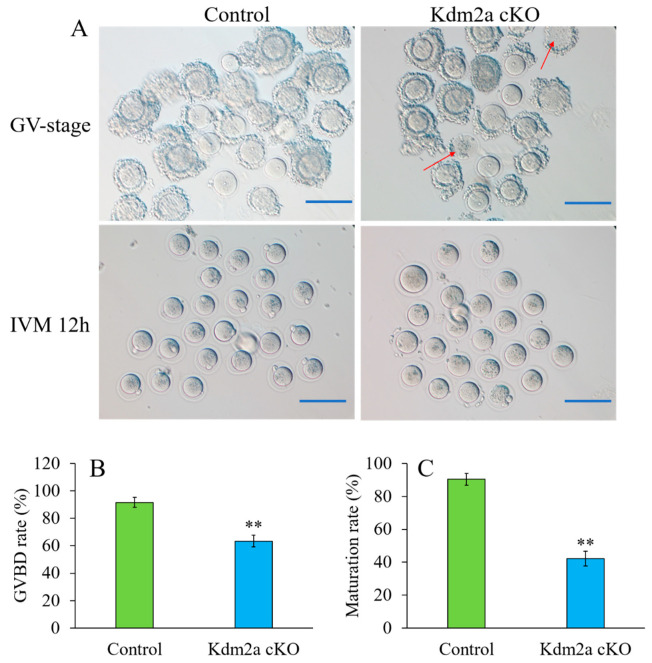
Oocytes from *Kdm2a* cKO mice were partially arrested to complete meiotic maturation. (**A**) Immature GV oocytes isolated from 4-week-old control and *Kdm2a* cKO mice were cultured in vitro to check the maturational progression, a few polar bodies can be excluded of *Kdm2a* cKO oocytes after 12 h culture. The arrow points to cytoplasmic damage. (**B**,**C**) Quantitative analysis of GVBD rate and Pb1 extrusion rate in control (n = 106) and *Kdm2a* cKO (n = 94) oocytes. The graph shows the mean ± SEM of the results obtained in three independent experiments. ** Extremely significant difference (*p* < 0.01). Scale bars, 200 μm.

**Figure 6 ijms-23-12008-f006:**
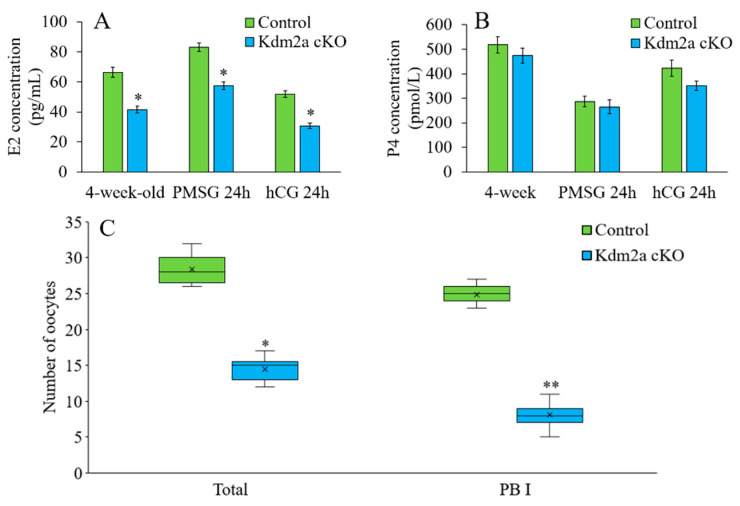
*Kdm2a* cKO mice exhibited lower serum estradiol without changing progesterone. (**A**) The concentrations of estradiol (E2) in the serum samples prepared from control and *Kdm2a* cKO mice at 4-weeks-old (n = 5), at 24 h after PMSG injection (n = 5), and 24 h after hCG injection (n = 5). (**B**) The concentrations of progesterone in the serum samples of control and *Kdm2a* cKO mice at 4-weeks-old (n = 5), at 24 h after PMSG injection (n = 5), and 24 h after hCG injection (n = 5). (**C**) The number of oocytes were ovulated after PMSG and hCG treatment. * Significant difference (*p* < 0.05), ** Extremely significant difference (*p* < 0.01).

**Figure 7 ijms-23-12008-f007:**
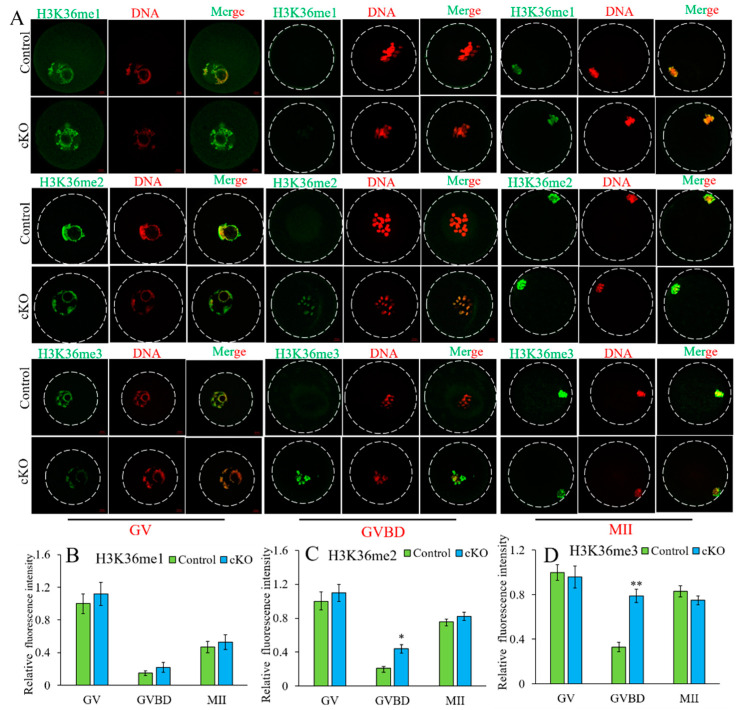
Analysis of H3K36me1/2/3 levels in the GV-, GVBD-, and MII-stage oocytes of control and *Kdm2a* cKO mice. (**A**) Analysis of H3K36me1/2/3 levels by IF. Oocytes were labeled with anti-H3K36me1/2/3 followed by appropriate Alexa Fluor-conjugated secondary antibodies (green) and were counterstained with the nuclear stain PI (red). (**B**–**D**) The H3K36me1/2/3 fluorescence intensity in oocytes of control and *Kdm2a* cKO mice by image pro-plus 6.0. * Significant difference (*p* < 0.05) and ** Extremely significant difference (*p* < 0.01) from the *Kdm2a* cKO group compared with the control group.

**Figure 8 ijms-23-12008-f008:**
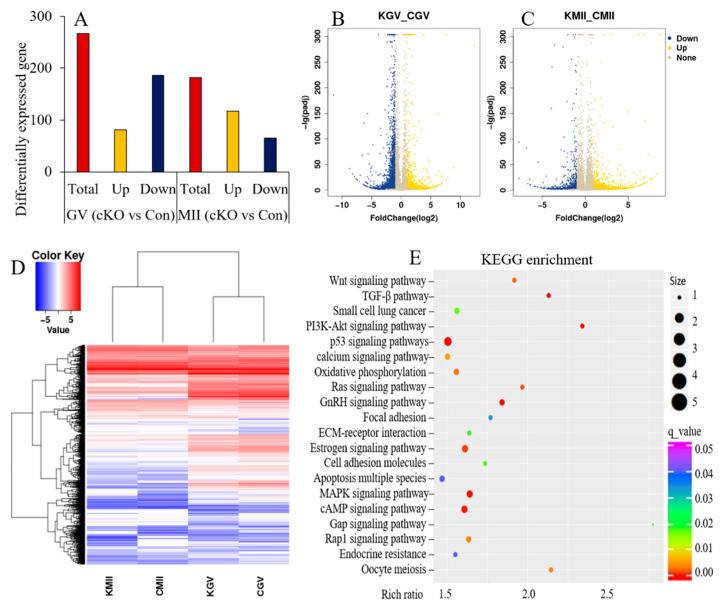
*Kdm2a* knockout impacts on the genes expression and signaling pathways during oocytes meiotic maturation. (**A**) the total number of differentially expressed genes between *Kdm2a* cKO (cKO) and control (Con) group oocytes of GV- and MII-stage, including up-regulated and down-regulated genes. (**B**) Volcano plot for the differentially expressed genes in GV-stage oocytes between *Kdm2a* cKO and control group, and (**C**) Volcano plot for the differentially expressed genes in MII-stage oocytes between *Kdm2a* cKO and control group. (**D**) Hierarchical clustering based on transcriptomic difference among GV-stage and MII-stage oocytes collected from *Kdm2a* cKO and control mice. (**E**) KEGG enrichment analysis of the differentially expressed genes in *Kdm2a* cKO oocytes compared with the control group. KGV: GV-stage oocytes from *Kdm2a* cKO group; CGV: GV-stage oocytes from control group; KMII: MII-stage oocytes from *Kdm2a* cKO group; CMII: MII-stage oocytes from control group.

**Figure 9 ijms-23-12008-f009:**
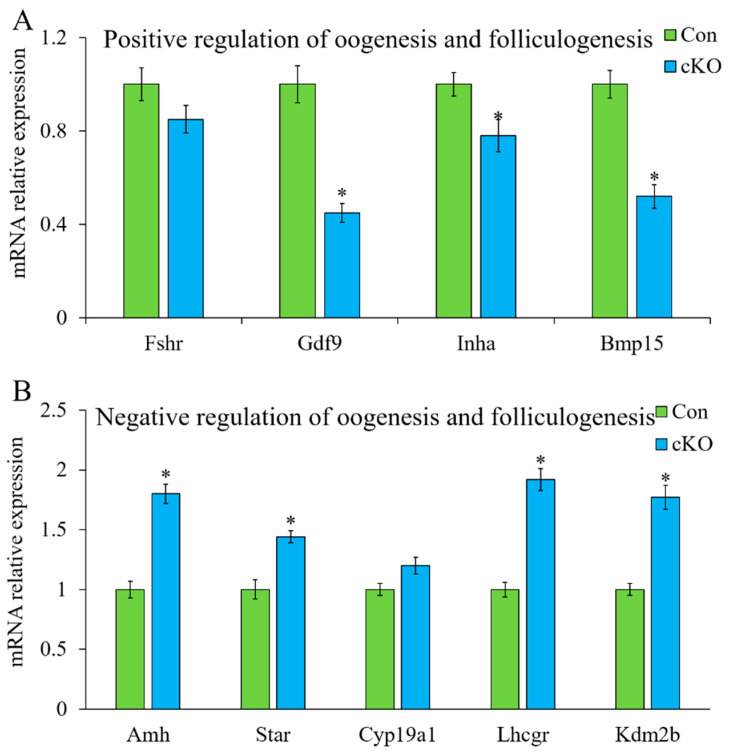
The mRNA expression levels of positive and negative genes related to regulate oogenesis and folliculogenesis in GV-stage oocytes. Four-week-old ovaries from *Kdm2a* cKO (cKO) and control (Con) mice were used to collect GV-stage oocytes; then, RT-qPCR was used to determine the levels of transcripts positively (**A**) and negatively (**B**) associated with oocyte and follicle development. n = 5 females/group. * Significant difference (*p* < 0.05).

## Data Availability

The data presented in this study are available on request from the corresponding another.

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
