# Peer review of "Oocyte-Specific Knockout of Histone Lysine Demethylase KDM2a Compromises Fertility by Blocking the Development of Follicles and Oocytes"

_ijms, 2022, doi:10.3390/ijms231912008_

Round 1

Reviewer 1 Report

Thank You for Your interesting work. The manuscript is well written and the methods and results sections are presented well. 

I have only some question to authors that need to be answered before publication of the manuscript. In my opinion some comment is needed in discussion section regarding the process of sexual maturation in mice. How the results in 4-, 6-, and 8-weeks mice pups can be extrapolated to human population? Are 4-, or 6-weeks pups ready to reproduce or they correspond to human children? At what age mice pups are too old to reproduce? If any of Your pups were ready to reproduce at any time-point (e.g 8-weeks pups), were the samples taken at a specific menstrual cycle phase (the identification of menstrual phase of rodents may be assessed on the number and proportion of cells observed in vaginal secretion)? Did You consider this as an important potential bias factor in the study? 

Author Response

Reviewer 1

I have only some questions to authors that need to be answered before publication of the manuscript. In my opinion some comment is needed in discussion section regarding the process of sexual maturation in mice. How the results in 4-, 6-, and 8-weeks mice pups can be extrapolated to human population? Are 4-, or 6-weeks pups ready to reproduce or they correspond to human children? At what age mice pups are too old to reproduce? If any of Your pups were ready to reproduce at any time-point (e.g 8-weeks pups), were the samples taken at a specific menstrual cycle phase (the identification of menstrual phase of rodents may be assessed on the number and proportion of cells observed in vaginal secretion)? Did You consider this as an important potential bias factor in the study? 

Answer: Thanks for your evaluation and suggestion.
First of all, 4-8 weeks after birth is the rapid growth period in mice, and vaginal opening of female mice in 4-week-old and start to generate sperm of male in 5-week-old, 6-week-old means sexual maturation in mice (Reference A). In this study, we chose 4-week-old (before sexual maturation), 6-week-old (sexual maturation) and 8-week-old (after sexual maturation) mice for comparative analysis to investigate the effects of Kdm2a on oocyte and follicular development.
Secondly, the results in 4-, 6-, and 8-weeks mice pups can be extrapolated to human population as the following figure (Reference B), 6-week-old female corresponds to human about 20 years old in female.
Third, the menstrual cycle is 4-5 days in mice, and being in different stages of the menstrual cycle has a certain impact on follicular developmental status and reproduction. Using mice in the same physiological period to do the experiments of reproduction is more accurate and reliable. In our study, the female of Kdm2a cKO (n=6) and Kdm2aflox/flox (n=6) were randomly selected without the identification of menstrual phase, and our reproductive experiment is no less than 4 months. To a certain extent, we thought that this factor could be ignored. 
Finally, we have added some comment in discussion section as “Although being in different stages of the menstrual cycle has a certain impact on reproduction, we inferred that this result was caused by the abnormal development of follicles and oocytes after Kdm2a deletion.”.

Thank you again for your suggestion, and we will pay more attention to this issue in the future work.

A. Flurkey, K., J. M. Currer, and D. E. Harrison. 2007.‘The mouse in biomedical research.’in James G. Fox (ed.), American College of Laboratory Animal Medicine series (Elsevier, AP: Amsterdam; Boston).

B. Aged C57BL/6J mice for research studies: considerations, applications, and best practices. 2020, The Jackson Laboratory.

Reviewer 2 Report

Comments on the manuscript:

“Oocyte-specific knockout of histone lysine demethylase KDM2a compromises fertility by blocking the development of follicles and oocytes”

Epigenetic factors are involved in gene expression profiles characteristic of cell types, without modifying DNA sequences. Histone methylation plays an important role in the development of follicles and oocytes. Lysine-specific demethylase 2a (KDM2a) is notably closely associated with gametogenesis and reproduction, according to regulatory mechanisms still poorly characterized in vivo.

To understand the role of histone lysine demethylase Kdm2a in follicles and oocytes, a conditional oocyte inactivation mouse transgenic model Kdm2a was generated and these mice were fertilized. After having verified Kdma deletion in model, visualized the presence of Kdma in oocytes and follicles in normal mice, the authors appreciated the effects of its deletion on fecundity and follicular development, the level of different molecules involved in reproduction and change in mRNA expression in oocytes.

This well-written study brings significant elements to the knowledge of epigenetic effects on the biology of reproduction, a topical issue. I think this article can be published after some improvements. Here are some remarks.

Page 1, lines 22-23. Abstract: “The abstract states “Our results showed that the number of pups was reduced by approximately 50%”, but how are the pups obtained? the abstract does not indicate the stage of fertilization of the females by the males, which interferes with the understanding of the study. This precision would be useful.

Page 1, Introduction. This study is a work of high technicality with many methods used to highlight the variations linked to the mutation. However, the purpose of the work does not seem sufficiently clear to me in the introduction. The latter ends with an appreciation concerning the study which is the first to demonstrate that the fact that Kdm2a is strongly expressed in the oocytes is essential for the development of oocytes and follicles. I think it would be useful to state the aim of the work in a few sentences.

Page 5, figure 3: a scalebar would be useful on figures 3A.

Page 7, figure 5: a scalebar would be useful on figures 5A.

Page 14, line 432: An inset showing negative controls obtained for example by omitting the first antibody would be helpful.

Page 15, RNA extraction and RT-qPCR, line 485: give a brief description as for the other techniques.

Page 15, line 587: is it ref 55 or 56 (I couldn't find ref 56 in the text)?

Author Response

Epigenetic factors are involved in gene expression profiles characteristic of cell types, without modifying DNA sequences. Histone methylation plays an important role in the development of follicles and oocytes. Lysine-specific demethylase 2a (KDM2a) is notably closely associated with gametogenesis and reproduction, according to regulatory mechanisms still poorly characterized in vivo.
To understand the role of histone lysine demethylase Kdm2a in follicles and oocytes, a conditional oocyte inactivation mouse transgenic model Kdm2a was generated and these mice were fertilized. After having verified Kdma deletion in model, visualized the presence of Kdma in oocytes and follicles in normal mice, the authors appreciated the effects of its deletion on fecundity and follicular development, the level of different molecules involved in reproduction and change in mRNA expression in oocytes.
This well-written study brings significant elements to the knowledge of epigenetic effects on the biology of reproduction, a topical issue. I think this article can be published after some improvements. Here are some remarks.
Answer: Thanks for your evaluation and suggestion, and your encouragement is the driving force for us to continue. 

Page 1, lines 22-23. Abstract: “The abstract states “Our results showed that the number of pups was reduced by approximately 50%”, but how are the pups obtained? the abstract does not indicate the stage of fertilization of the females by the males, which interferes with the understanding of the study. This precision would be useful.
Answer: Thanks for your suggestion, and we have revised as “Our results showed that the number of pups was reduced by approximately 50% in adult Kdm2a cKO female mice mating with wildtype male than that of the control (Kdm2aflox/flox) group.”.

Page 1, Introduction. This study is a work of high technicality with many methods used to highlight the variations linked to the mutation. However, the purpose of the work does not seem sufficiently clear to me in the introduction. The latter ends with an appreciation concerning the study which is the first to demonstrate that the fact that Kdm2a is strongly expressed in the oocytes is essential for the development of oocytes and follicles. I think it would be useful to state the aim of the work in a few sentences.
Answer: Thanks for your suggestion. We have added “Thus, the aim of the present study is to investigate the effects of Kdm2a on the oocyte growth and female fertility, and to explore the potential mechanism.” in this part.

Page 5, figure 3: a scalebar would be useful on figures 3A.
Answer: Thanks for your suggestion. We have added the scalebar.

Page 7, figure 5: a scalebar would be useful on figures 5A.
Answer: Thanks for your suggestion. We have added the scalebar.

Page 14, line 432: An inset showing negative controls obtained for example by omitting the first antibody would be helpful.
Answer: Thanks for your suggestion. We have revised here as “The negative group incubated with PBS in place of the primary antibody.”, and supplied the negative images in Figure 2.

Page 15, RNA extraction and RT-qPCR, line 485: give a brief description as for the other techniques.
Answer: Thanks for your suggestion. We have revised as “RNA extraction from oocytes (20 oocytes/sample) using the Cells-to-SignalTM Kit (Invitrogen, USA) followed the protocol of the manufacturer as previously described [54]. Briefly, each group of oocytes were washed twice with PBS and transferred to 10 μL Lysis Buffer. The sample was gently shaken for 3 min to lyse the oocytes. The isolated RNA was immediately used for reverse transcription with the cDNA synthesis kit (Takara, China) according to the manufacturer’s instructions, and then used as templates for PCR analysis.”.

Page 15, line 587: is it ref 55 or 56 (I couldn't find ref 56 in the text)?
Answer: Thanks for your suggestion. Page 15 line 487 is ref 55, not 56. Ref 56 is appeared on Page 15 line 500 of “4.9. Western blot”. Furthermore, we have checked all the references carefully in the full text.
